# Proof of concept for real-time detection of SARS CoV-2 infection with an electronic nose

**Kobi Snitz[1]☯\*, Michal Andelman-Gur[1]☯, Liron Pinchover[1], Reut Weissgross[1], Aharon Weissbrod[1], Eva Mishor[1], Roni Zoller[1], Vera Linetsky[1], Abebe Medhanie[1], Sagit Shushan[1,2], Eli Jaffe[3], Noam Sobel[ID][1]\***

**1** Department of Neurobiology and Azrieli Center for Human Brain Imaging and Research, Weizmann Institute of Science, Rehovot, Israel, **2** Department of Otolaryngology & Head and Neck Surgery, Edith Wolfson Medical Center, Holon, Israel, **3** Magen David Adom in Israel and Department of Emergency Medicine, Ben-Gurion University of the Negev, Beer Sheva, Israel

☯ These authors contributed equally to this work.
\* noam.sobel@weizmann.ac.il (NS); kobi.snitz@weizmann.ac.il (KS)

## Abstract

Rapid diagnosis is key to curtailing the Covid-19 pandemic. One path to such rapid diagnosis may rely on identifying volatile organic compounds (VOCs) emitted by the infected body, or in other words, identifying the smell of the infection. Consistent with this rationale, dogs can use their nose to identify Covid-19 patients. Given the scale of the pandemic, however, animal deployment is a challenging solution. In contrast, electronic noses (eNoses) are machines aimed at mimicking animal olfaction, and these can be deployed at scale. To test the hypothesis that SARS CoV-2 infection is associated with a body-odor detectable by an eNose, we placed a generic eNose in-line at a drive-through testing station. We applied a deep learning classifier to the eNose measurements, and achieved real-time detection of SARS CoV-2 infection at a level significantly better than chance, for both symptomatic and non-symptomatic participants. This proof of concept with a generic eNose implies that an optimized eNose may allow effective real-time diagnosis, which would provide for extensive relief in the Covid-19 pandemic.

## Introduction

Viruses alone don't produce volatile organic compounds (VOCs), but virus-infected cells do [1], and these can be targeted for VOC-based disease detection [2]. Such detection can be conducted with trained animals such as dogs [3, 4], but large-scale animal deployment is a challenge. In turn, electronic noses (eNoses) are machines that mimic the animal olfactory system [5–7], and these can be deployed at scale. eNoses typically contain an array of sensors, each optimized for a different chemical range [8], and the readout of their resultant multi-sensor pattern can be "trained" to identify targets ranging from viral or bacterial infections [9–13], to non-infectious diseases [14, 15].

In an ideal setting, one would first use analytical equipment such as gas-chromatography mass-spectrometry (GCMS) to identify the VOCs of interest, and then optimize the sensors within the eNose (typically by selectively coating them) so as to best detect those target VOCs

**Data Availability Statement:** All relevant data are within the manuscript and its Supporting Information files.

**Funding:** This study was supported by pilot grants from MAFAT: The Israeli Ministry of Defense Directorate of Defense Research and Development, and from Sonia T. Marschak. This effort also relied on ongoing Sobel lab resources, provided by a European Research Council AdG. grant #670798 (SocioSmell) and Horizon 2020 FET Open project #662629 (NanoSmell). The funders had no role in study design, data collection and analysis, decision to publish, or preparation of the manuscript.

**Competing interests:** The authors declare no competing interests.

[5]. The problem with this ideal path is that it takes time, and time is one thing we don't have in the Covid-19 pandemic. Given initial data suggesting dogs may be able to smell Covid-19 patients [16], it has been suggested that using eNoses to do the same may provide for a much-needed aid in the fight against the Covid-19 pandemic [17]. With this in mind, we set out to test whether a generic eNose could be used to detect SARS CoV-2 infection. Aiming for application in a real-world setting, we were faced with deciding what body-odor source to sample. Much of the eNose diagnostics effort in the literature is focused on exhaled breath analysis. Such breath sampling and analysis has standard protocols [18, 19], and has reached at achievements in cases such as identifying pneumonia [20], tuberculosis [11, 21], asthma and COPD [22], respiratory infections [23] and lung malignancies [24–26], and recently indeed for COVID-19 [27, 28]. Moreover, eNose measurements of exhaled breath may inform on non-respiratory conditions as well, such as neurodegenerative illnesses [15]. Here, however, we opted not to target exhaled breath analysis per se. Rather, we observe that the nasal passage has been implicated as a site of SARS CoV-2 infection [29, 30]. Therefore, our goal is to "smell" the inner nasal passage itself. From our perspective, breath, and its associated lung-derived VOCs, are an inevitable source of noise in the nasal passage, but not our intended target. Thus, we set out to develop methods that differ from the standard breath sampling and analysis typically applied in the field. Moreover, we then applied these methods in a real-world uncontrolled field environment. We found that despite various sources of noise, we could detect SARS CoV-2 infection at above chance levels. This implies there is a signal in this source, but only if optimized in the future, may this approach have clinical value.

## Methods

### Participants

We placed our experiment in-line at a national testing station ran by Magen David Adom, the Israeli equivalent of the Red Cross, in Tel Aviv, Israel. People were sent to the station by a national referring system that assigned tests to individuals who had a lengthy exposure to a verified Covid-19 patient, or were experiencing persistent Covid-19 symptoms. These were the only selection criteria applied. We further excluded minors, as we did not have ethical approval to include them in our study. With these criteria, we tested 503 individuals (229F, 267M, 4 unidentified sex, mean age 38.9), of which 27 (12F, mean age 35.85) were later deemed SARS CoV-2 positive (5.4%) by RT-PCR. RT-PCR was conducted in one of several nationally certified labs, and we followed up all positive diagnoses with phone-interviews to verify the result and again verify the lack or presence of symptoms. All participants provided informed written consent to procedures approved by the Weizmann Institute of Science Institutional Review Board (IRB). The individuals pictured in Figs 1 and 2 and in S1 Video have provided written informed consent (as outlined in PLOS consent form) to publish their image alongside the manuscript.

### Electronic nose set-up

We used a PEN3 eNose (AIRSENSE Analytics GmbH, Schwerin, Germany). The PEN3 is a compact (92 × 190 × 255 mm) lightweight (2.3 kg) device, consisting of a gas sampling unit and a sensor array. The sensor array is composed of 10 different thermo-regulated metal oxide sensors, positioned in a stainless-steel chamber (volume: 1.8 ml, temperature: 110˚C). Each sensor is uniquely coated, rendering it particularly sensitive to a restricted class of chemical compounds. When a compound interacts with the sensor, this results in an oxygen exchange that leads to a change in electrical conductivity [31]. The specific sensitivities of the sensors are in Table 1. We used the PEN3 with its native sampling software (WinMuster), and the

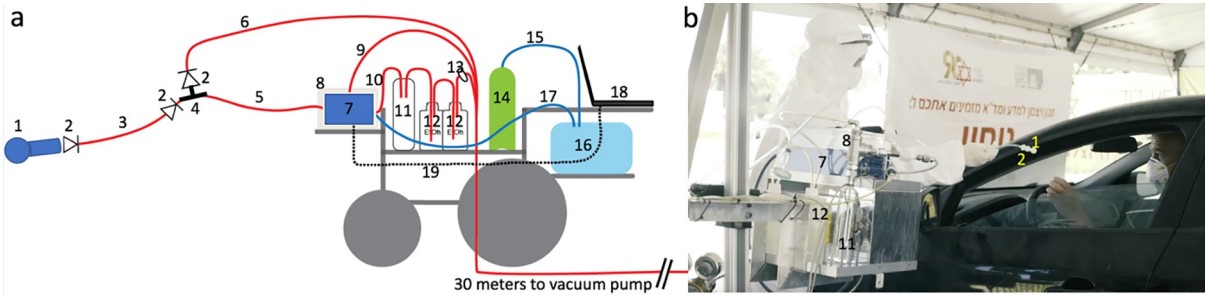

**Fig 1. A mobile eNose platform was deployed at a drive-through testing station. A.** The detailed logic of this set-up is described in the Methods under "electronic nose set-up". Components not drawn to scale. 1. The individual disposable sampling valve. 2. One-way flow valve. 3. 1-meter long disposable Tygon tube. 4. The T junction with quick-connect. 5. The eNose inlet tube. 6. The cleaning overflow pull away line. 7. The PEN 3 eNose. 8. Plexiglass mini-hood. 9. Vacuum line pulling from hood. 10. eNose exhaust line. 11. Liquid trap. 12. Ethanol canisters. 13. Filter. 14. Medical grade breathable air canister. 15. Clean air line into air bag. 16. Air bag. 17. Clean reference air inlet to eNose. 18. Laptop. 19. USB line from laptop to eNose. **B.** The image is a screenshot from the S1 Video, depicting the experimenter handing a sampling valve to a participant. Visible system components numbered as in A. The person in the car is a co-author demonstrating, and not a participant, and informed consent for publication of identifying images in an online open-access journal was obtained.

following settings: Chamber flow = 400ml/min, Flush time = 40s, Zero-point trim time = 10s, Measurement time = 80s. In the following paragraph, numbers in parenthesis relate to the numbered elements in Fig 1. For the current experiment we placed the entire sensing

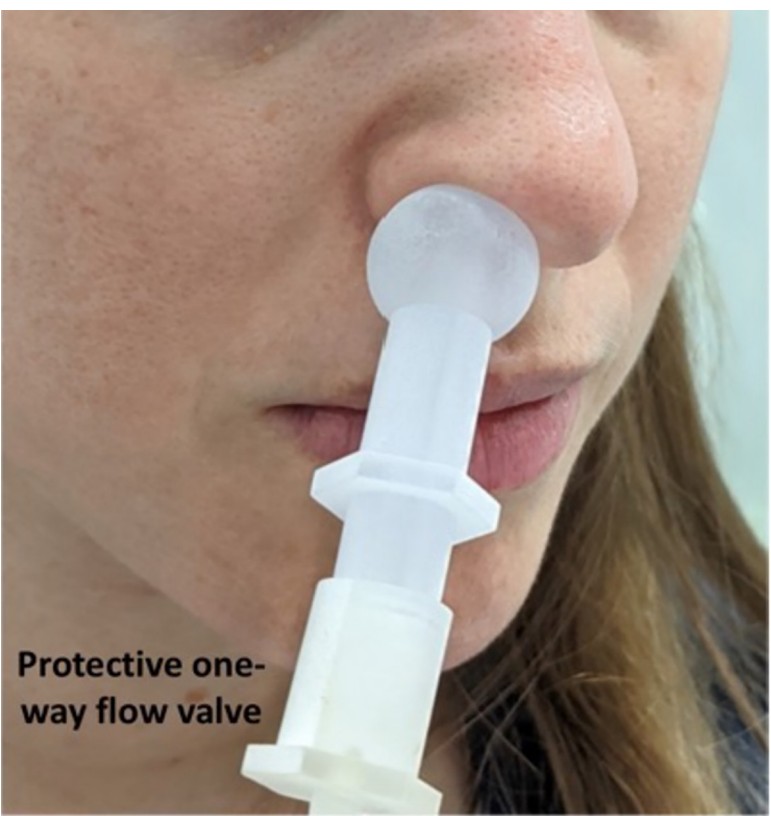

**Fig 2. A one-way disposable sampling valve protected participants.** The sampling valve fits snugly against the nostril, providing an air-tight connection for pulling air from within the nostril, independent of exhalation. The person in the image is a co-author demonstrating, and not a participant, and informed consent for publication of identifying images in an online open-access journal was obtained.

**Table 1. eNose sensor functionalization.**

| Sensor number | Sensor name | Object substances for sensing | Limit of detection |
|---|---|---|---|
| Sensor 1 | W1C | Aromatics | 5 ppm |
| Sensor 2 | W5S | Ammonia and aromatic molecules | 1 ppm |
| Sensor 3 | W3C | Broad-nitrogen oxide | 5 ppm |
| Sensor 4 | W6S | Hydrogen | 5 ppm |
| Sensor 5 | W5C | Methane, propane, and aliphatics | 1 ppm |
| Sensor 6 | W1S | Broad-methane | 5 ppm |
| Sensor 7 | W1W | Sulfur-containing organics | 0.1 ppm |
| Sensor 8 | W2S | Broad-alcohols, broad-carbon chains | 5 ppm |
| Sensor 9 | W2W | Aromatics, sulfur- and chlorine-containing organics | 1 ppm |
| Sensor 10 | W3S | Methane and aliphatics | 5 ppm |

The specific functionalization of each of the 10 sensors in the PEN3 eNose as defined by the manufacturer.

apparatus on the chassis of an electric wheelchair, to as to provide for system mobility. At the front of the wheelchair we secured an electric lift (FA-200-TR-24-60, Firgelli, Laverton North, Australia), and placed the eNose (Fig 1, #7) on the lift shelf. This allowed adjusting eNose height to individual car window height. On the wheelchair we also secured a canister of pressurized medical-grade breathing air (Fig 1, #14). This air was used to continuously inflate a large breathing bag (Xenon-133 Rebreathing System, Biodex, Shirley NY, USA) (Fig 1, #16), and this served as the reference air source for the eNose. This arrangement assured a consistent reference regardless of any environmental changes that may have occurred at the testing station over time. Finally, we took several steps to assure the safety of both the experimenters and participants, and protect them from the risk of infection from our system. The PEN3 has a sample exhaust port at its back. We directed this potentially infected exhaust through a one-way flow valve into a liquid trap (to protect the eNose from back-flow) (Fig 1, #11) and then directly into the bottom of a 1-liter 70% ethanol canister (Fig 1, #12). The exhausted air bubbled through the ethanol, and continued into the bottom of a second 1-liter 70% ethanol canister (Fig 1, #12). After the exhausted air bubbled through the second canister, it passed through a glass-microfiber filter (GasVent 2000 01, GVS, Morecambe, UK) (Fig 1, #13), and was directly vacuumed to a pump situated about ~30 meters outside of the testing station tent. There the exhausted air passed through an additional filter, before mixing with the outside environment. This flow path promised that air sampled from the participants was both treated and distanced. In addition, we considered the unlikely possibility that the PEN3 might have some internal leak. To address this remote possibility, we enclosed the entire device in a plexiglass box (Fig 1, #8), that had a 1/2-inch tube (Fig 1, #9) pulling ~30 LPM from the box top to the same distant location, ~30 meters outside of the testing station tent. In other words, the PEN3 was within a closed mini-hood. Moreover, in its cleaning phase, the PEN3 can push overflow air out through its inlet port. To address this possible source of contamination, we placed a "T" at the tip of the inlet tube (Fig 1, #4), with two all-Teflon ¼-inch one-way flow valves (CV-4-4-P-05, iPolymer, Irvine, CA, USA) (Fig 1, #2). One valve prevented flow from the device towards the participant, and the other valve directed such overflow cleaning air into the same line that led ~30 meters outside of the testing station tent. Finally, at the "T" we had a quick-connector, to which we attached the disposable unit used for each participant. This disposable unit included one-meter long ¼-inch Tygon tubing (Fig 1, #3), ending at a sampling valve (Fig 1, #1, and Fig 2). This 3D printed valve was shaped so as to fit snugly against the nostril from which it pulled air. This valve also contained a final one-way flow valve, so that if

somehow something went wrong at the eNose apparatus, perhaps a breakdown following extended use, each participant was in this way nevertheless behind an added individual layer of protection (Fig 2). This entire system was reviewed by the device safety unit at the Israeli Ministry of Health, and was granted safety approval.

## Procedures

Cars were typically queued up at the testing station. An experimenter in full personal protective equipment (PPE) approached the car, and through a slightly open window explained the purpose of the experiment and its procedures, and requested participation. If the person in the car agreed, he/she was handed informed consent documents that included spaces for their Israeli national ID number (this number is used by the health system for obtaining results), their phone number, questions on age, sex, and whether the person was experiencing symptoms (yes/no without details). In an effort to minimize the transfer of potentially infected materials back to the experimenters, we did not collect the informed consent documents, but rather photographed the signature and information page through the car window. The participant was then handed the sampling valve, and instructed to hold it snugly against a nostril opening for 80 seconds. The shape of the sampling valve (Fig 2) ensured an air-tight application to the nose, such that outside air was not directly sampled. The participants were told to breath normally, but only through their open mouth, during these 80 seconds. After the sample, the disposable sampling valve and tube were discarded, and the participants advanced about 10 meters to the RT-PCR swabbing station. In other words, this experiment, by design, was double-blind.

## Analysis

All analyses were conducted using Matlab software (Mathworks, USA). We applied the long short-term memory (LSTM) deep-learning classifier algorithm [32] to the entire time-series. We then applied a leave-one-out cross validation: We randomly selected 26 out of 27 positive samples, trained the classifier on these 26 positive samples, and then tested whether it can select between the one remaining positive sample versus one randomly selected negative sample (Fig 3). Chance performance at this task is 50%. We repeated this process 500 times, each time testing a different sample, in order to obtain a true-positive rate. Finally, because no selection of 500 is privileged, we repeated this entire process 100 times in order to obtain a distribution of possible true-positive rates. To estimate the statistical significance of the outcome, we

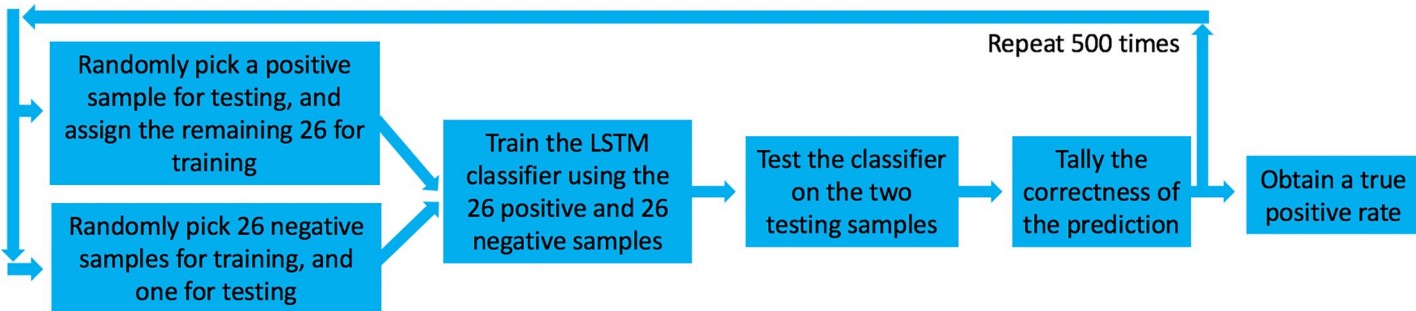

**Fig 3. Analysis path.** We randomly selected 26 out of 27 positive samples, trained the classifier on these 26 positive samples, and then tested whether it can select between the one remaining positive sample versus one randomly selected negative sample. Chance performance at this task is 50%. We repeated this process 500 times, each time testing a different sample, in order to obtain a true-positive rate. Finally, because no selection of 500 is privileged, we repeated this entire process 100 times in order to obtain a distribution of possible true-positive rates.

repeated the entire process 600 times, but here first arbitrarily deemed a random 27 negative samples as "positive". In this manner, we can ask what is the chance probability of obtaining a similar result. The entire analysis code, with explanatory comments, is in the S1 Materials folder, that contains all the raw data, and all the code used in the analysis of this manuscript.

## Results

We successfully deployed at a drive-through testing station [33] in Tel Aviv, Israel (Fig 1B, S1 Video). Here, individuals being tested never exited their car, but rather drove through one of several lanes housing a testing team that swabbed them through an open window for later analysis by reverse-transcription polymerase chain reaction (RT-PCR) test for pathogen identification [34]. Individuals were referred to the testing station only if they had exposure to a verified Covid-19 patient, or were persistently experiencing Covid-19 symptoms. We placed our testing station in line with one such lane, directly before swabbing, and offered drivers to participate in our experiment. After providing written informed consent to procedures carried out in accordance with relevant guidelines and regulations, they were handed a disposable sampling valve (object #1 in Figs 1 and 2) that was linked to the eNose inlet port by a disposable flexible tube. They were instructed to secure this sampling valve air-tight to their nostril for 80 seconds while the eNose pulled sampled air at 400cc per minute (S1 Video). They then continued directly to RT-PCR testing.

Compliance was very high, with about 81% agreeing to participate. In 22 days of deployment, we obtained 503 samples (229F, 267M, 4 unreported sex, mean age 38.9), of which 27 (12F, 15M, mean age 35.85) were later deemed SARS CoV-2 positive (5.4%) by RT-PCR.

To ask whether we could use the eNose to identify SARS CoV-2 infection, we enacted the following analysis scheme: Initially, we visually inspected the data and observed that sensors #1, #3, and #5 never responded to any sample (Fig 4A). We therefore discarded these sensors from further analysis (this information may help guide a future search for COVID-19 related VOCs, considering the sensitivity specification of each sensor [31]). Next, we observe that PEN3 sensors each have a dynamic response pattern over about 65 seconds from sampling onset, before reaching equilibrium (Fig 4A). To ask whether relevant information was either in the entire time-series, or at the point of equilibrium alone, we conducted two principal component analyses (PCA) [35]: one using equilibrium end-values alone, and the other using a 3rd

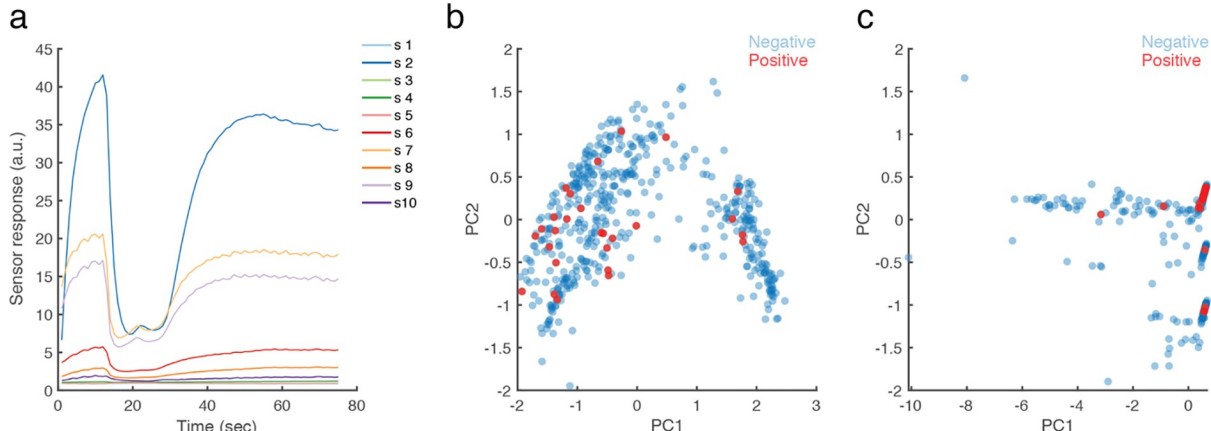

**Fig 4. eNose measurements cluster SARS CoV-2 positive participants. A**. An example of the raw eNose measurement from one participant. Each line is one of 10 sensors. **B**. PCA of the equilibrium end-values only. **C**. PCA of the 3rd degree polynomial fit of the entire time-series. n = 503. The 27 positive samples are in red.

degree polynomial fit of the entire time-series. We observe no evidence for clustering of positive and negative test results in the equilibrium-value PCA (Fig 4B), but evidence for clusters in the time-series PCA (Fig 4C).

With this encouraging observation in hand, we applied the long short-term memory (LSTM) deep-learning classifier algorithm [18] to the entire time-series. We then applied a leave-one-out cross validation: We randomly selected 26 out of 27 positive samples, trained the classifier on these 26 positive samples, and then tested whether it can select between the one remaining positive sample versus one randomly selected negative sample (Fig 3). Chance performance at this task is 50%. We repeated this process 500 times, each time testing a different sample, in order to obtain a true-positive rate. Finally, because no selection of 500 is privileged, we repeated this entire process 100 times in order to obtain a distribution of possible true-positive rates. We observe a true positive rate ranging between 61% and 71%, with mean at 66.7% (Std Dev = 2%) (Fig 5A in red). The associated mean false negative rate was 33.3%, and the mean false positive rate was 57% (Std Dev = 2%) (Fig 5B in yellow). This combines for a receiver operating characteristic curve (ROC) that is modestly but consistently above the diagonal line, with an area under the curve (AUC) of 0.58 (Fig 5C). In relation to results with RT-PCR [36], the 66.7% true positive rate may be seen as promising, but considering the shallow ROC curve, is it significantly different from 50% chance? To assess this, we repeated the entire above process, but now first randomly selected 27 negative samples, and arbitrarily labeled them as the positive samples. We repeated this 600 times, and observe a true positive rate that indeed centered at 50% (Std Dev = 7%), and only 11 times in 600 repetitions reached 66.7% (Fig 5A in blue). This assigns a Bootstrap p value of less than 0.02 to our result (Hedges' g effect size = 2.48). Moreover, a post-hoc analysis of power implied that at 27 positive samples, this result has 94% power (Fig 5D, Supplementary Fig 1 in S1 File). Finally, could have we been detecting general malaise rather than SARS CoV-2 infection? We observe that of the 27 positive participants, 14 were non-symptomatic. We therefore repeated the entire analysis scheme, now using only these 14 non-symptomatic positive samples instead of the 27 samples. We obtained a mean ROC AUC of 0.63, making for a true positive rate ranging from 47.4% to 94.4%, with mean at 75.8% (Std Dev = 12%) (we disregarded 2 outliers out of the 100, who were at 16.4% and 33.4%, i.e., more than 3 Std Dev from the mean) (Fig 5E in red). To test whether this is significantly different from chance, we again arbitrarily assign 14 negative sample as positive, and repeat the analysis 600 times. In only 34 cases did we obtain a similar outcome, thus assigning a Bootstrap p value of 0.057 to this result (Hedges' g effect size = 2.0) (Fig 5E in blue). We conclude that it is unlikely that we were merely detecting general malaise rather than SARS CoV-2 infection.

## Discussion

This manuscript had a modest goal: to test the hypothesis that there is a body-odor, and more specifically an intra-nasal-passage body-odor, associated with SARS CoV-2 infection detectable by eNose. To this we provide a positive answer, providing proof of concept for VOC-based real-time detection of SARS CoV-2 infection. We note that results to this effect have also recently been obtained by others: In one study [28], the authors used a commercial metal-oxide-sensor-based eNose, albeit different from ours (Aeonose, The Aeonose Company, Zutphen, the Netherlands). They sampled oral exhaled breath from 219 participants, of which 57 were COVID-19 positive. These participants were tested at two locations, all the negative participants were tested at an outpatient clinic, and most all of the positive patients were tested at a COVID-19 nursing unit. This group obtained an ROC AUC of 0.74, compared to our ROC AUCs of 0.58 and 0.63. In a second study [27], the authors used a proprietary nanomaterial-

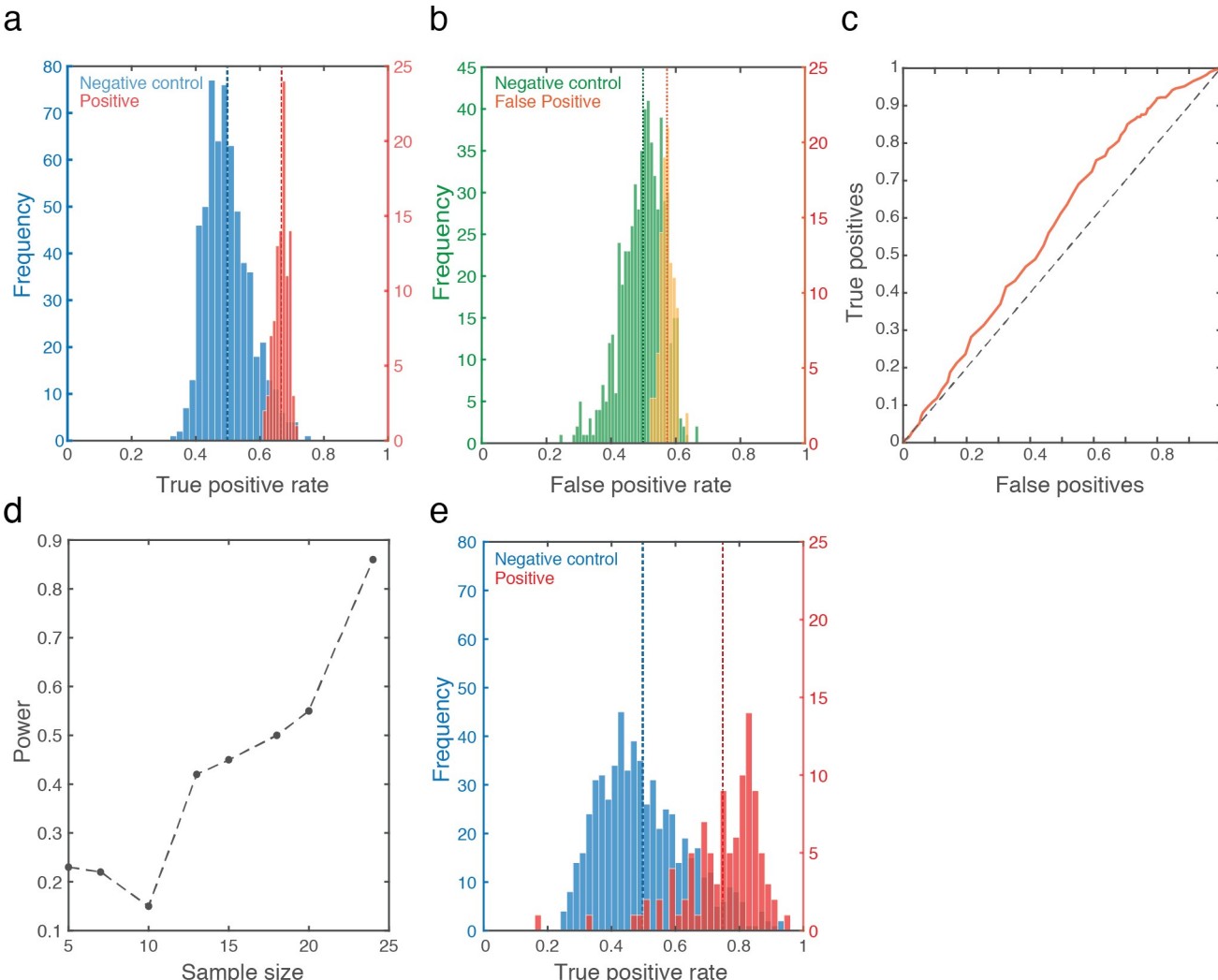

**Fig 5. An eNose can smell SARS CoV-2 infection. A.** In red, histogram of 100 true-positive values generated by the classifier (each value is the result of 500 selections of one of 27 positive and one of 476 negative), with the mean success rate (66.7%) in dashed red line. In blue, the control analysis: Histogram of 600 true-positive values generated by the classifier (each value is the result of 500 selections of one of 27 negatives now randomly assigned as positive, and one of 449 remaining negatives), with the mean (50%) in dashed blue line. **B**. The same analysis as in A, here depicting false positives (yellow) and random control (green). **C.** ROC curve. **D.** Power analysis on increasing sample sizes (see Supplementary Fig 1 in S1 File). **E.** Same as A, but using only the 14 non-symptomatic positive participants. Note two discarded outliers at 16.4% and 33.4%.

based hybrid sensor array eNose. This group also sampled oral exhaled breath, from 140 participants, of which 49 were COVID-19 positive, 58 were COVID-19 negative, and 33 were other lung disease participants. These participants were tested at multiple locations, with all the COVID-19 positive participants coming from the same intensive care unit at a hospital in Wuhan, China. This group obtained an ROC AUC of 0.81, again compared to our ROC AUCs of 0.58 and 0.63. Thus, the two current published efforts that we are aware of obtained results significantly better than ours. What underlies these differences? The likely explanation is that these groups used a better device, and applied it using better sampling methods. We do not say this facetiously. In turn, we would like to highlight one strength of our effort that we think is meaningful: We sampled in a naturalistic setting where, unlike in the two above examples, sources of variance where equally distributed across groups. A limitation of eNose diagnostics

is that we often don't know the precise molecular identity of the signal. Thus, had we tested a homogenous hospitalized cohort, did we then detect COVID-19, or did we detect the food/ laundry-detergent/any other commonality of the jointly housed COVID-19 positive participants? Here, by going directly to the very difficult end-test, namely the extremely noisy (olfaction-wise) and highly variable environment of the drive-through testing station, we provide a powerful inherently double-blind proof of concept for our hypothesis. Here, one SARS CoV-2 positive person may have just eaten a tuna-fish sandwich, another may have just brushed their teeth, and a third may have just doused themselves with their favorite perfume. Unlike an olfaction-wise homogenous group of patients in a hospital (who eat the same food, wear commonly laundered clothing, etc.), the only thing these people have in common is a later-obtained SARS CoV-2 positive test. With this noise in mind, we nevertheless obtain statistically significant classification. In contrast to this strength, our study has several weaknesses. First, our sampling strategies clearly introduce even added noise. For example, air from the car interior can enter through the unsampled nostril and mouth, and make its way to the sampling apparatus. Beyond this, the primary weakness that limits our effort to the status of proof of concept alone is the level of false positives. Although as noted our statistically significant 66.7% true positive rate (or 75.8% for non-symptomatic) is not far off of RT-PCR true positive rates [28], and we obtain this result instantaneously rather than days later, our 57% false positive rate prevents this method from deployment in its current form. Thus, our result is a basic-science proof of concept, and not a clinical tool. We think, however, that this shortcoming is technical rather than conceptual, and we have identified several steps towards addressing this limitation. For example, a major source of olfactory noise in our apparatus was the individual sampling valve that we used (Fig 2). Its manufacturing process included a disinfection procedure that introduced significant added noise to our measurements. Additional improvements may also include optimization of sensor coating based on analysis of Covid-19 volatiles. Given our current results with a generic eNose, we speculate that an optimized eNose may be able to provide effective real-time diagnoses in locations such as airports, the work-place, and cultural events, and in this potentially contribute to social and economic recovery in the COVID-19 pandemic.

## Supporting information

**S1 Video. A video demonstrating activity at the drive-through.** The person being tested is an experimenter demonstrating, and not a participant.
(MP4)

**S1 Materials. A folder containing all the raw data, the code, and a readme file explaining how to run it.**
(ZIP)

**S1 File.**
(PDF)

## Acknowledgments

We thank Ofer Perl for photography in Fig 2. We thank Shoval Silbert and Stratasys for their gracious help at 3D printing, Dr. Nadav Sheffer at the Ministry of Health, and Miki Segal, Efi Levav, and Ilan Klein from MDA for their gracious hospitality and help at the drive-through station.

## Author Contributions

**Conceptualization:** Kobi Snitz, Michal Andelman-Gur, Liron Pinchover, Sagit Shushan, Eli Jaffe, Noam Sobel.

**Data curation:** Kobi Snitz, Michal Andelman-Gur, Liron Pinchover, Noam Sobel.

**Formal analysis:** Kobi Snitz, Michal Andelman-Gur, Roni Zoller, Noam Sobel.

**Funding acquisition:** Noam Sobel.

**Investigation:** Kobi Snitz, Michal Andelman-Gur, Liron Pinchover, Reut Weissgross, Aharon Weissbrod, Roni Zoller, Vera Linetsky, Abebe Medhanie, Noam Sobel.

**Methodology:** Kobi Snitz, Michal Andelman-Gur, Liron Pinchover, Reut Weissgross, Aharon Weissbrod, Sagit Shushan, Noam Sobel.

**Project administration:** Liron Pinchover, Reut Weissgross, Aharon Weissbrod, Eli Jaffe, Noam Sobel.

**Resources:** Eli Jaffe, Noam Sobel.

**Supervision:** Sagit Shushan, Noam Sobel.

**Validation:** Noam Sobel.

**Visualization:** Eva Mishor, Noam Sobel.

**Writing – original draft:** Michal Andelman-Gur, Noam Sobel.

**Writing – review & editing:** Kobi Snitz, Michal Andelman-Gur, Liron Pinchover, Reut Weissgross, Aharon Weissbrod, Eva Mishor, Roni Zoller, Sagit Shushan, Eli Jaffe, Noam Sobel.

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
