## [Decision Letter · Decision Letter 0]

26 Oct 2020

PONE-D-20-27895

Proof of Concept for Real-Time Detection of SARS CoV-2 Infection with an Electronic Nose

PLOS ONE

Dear Dr. Sobel,

Thank you for submitting your manuscript to PLOS ONE. After careful consideration, we feel that it has merit but does not fully meet PLOS ONE’s publication criteria as it currently stands. Therefore, we invite you to submit a revised version of the manuscript that addresses the points raised during the review process.

The reviewers raised several issues that must be addressed to validate the methodology behind your work:

1. Please answer the concerns of reviewer #2 about the necessity to use of a sealed mask to prevent the detection of environmental contaminants, as well as the possibility that "normal" breathing patterns might not be adequate to detect endogenous VOCs. 

2. Please clarify the discrepancy between the expectation of reviewer #2 that the sensor’s conductivity should increase with the exhaled gas concentration whereas you appear to be observing the opposite. 

3. As requested by reviewer #1, it would be important to report any information you might have about the parameters of exhaled breath (humidity) affecting the sensor's readout.

In your revised manuscript, you're encouraged (i) to add to your introduction a review of published applications of the electronic nose technology to the diagnosis of lung disease; (ii) to include a discussion of any specific biomarkers you found associated with the VOCs of SARS CoV-2.

We look forward to receiving your revised manuscript.

Kind regards,

Matthieu Louis

Academic Editor

PLOS ONE

Journal Requirements:

2. We note that Figure 1 and Online Video SnitzMovie includes images of participants in the study. 

3.  We note that you have a patent relating to material pertinent to this article. Please provide an amended statement of Competing Interests to declare this patent (with details including name and number), along with any other relevant declarations relating to employment, consultancy, patents, products in development or modified products etc. Please confirm that this does not alter your adherence to all PLOS ONE policies on sharing data and materials, as detailed online in our guide for authors http://journals.plos.org/plosone/s/competing-interests by including the following statement: "This does not alter our adherence to  PLOS ONE policies on sharing data and materials.” If there are restrictions on sharing of data and/or materials, please state these. Please note that we cannot proceed with consideration of your article until this information has been declared.

Reviewers' comments:

Reviewer's Responses to Questions

**Comments to the Author**

1. Is the manuscript technically sound, and do the data support the conclusions?

Reviewer #1: Yes

Reviewer #2: No

2. Has the statistical analysis been performed appropriately and rigorously? 

Reviewer #1: Yes

Reviewer #2: No

3. Have the authors made all data underlying the findings in their manuscript fully available?

Reviewer #1: Yes

Reviewer #2: No

4. Is the manuscript presented in an intelligible fashion and written in standard English?

Reviewer #1: Yes

Reviewer #2: Yes

5. Review Comments to the Author

Reviewer #1: The authors present a proof-of-concept of SARS CoV-2 detection with a commercially available e-nose. They present a very detailed description of in-line sampling methods at a national testing station in Tel Aviv, Israel. Many details are provided on the strategy deployed to operate in a safe environment and avoid contamination during the test of each volunteer. This is valuable contribution for future studies aimed to develop and test e-noses for the detection of such infection.

The data analysis was carried out correctly with machine-learning methods. The authors also provide a critical assessment of points of strength and weaknesses of both sampling method and data analysis and recognize that the main limit of their method is the level of false positives, which hinders the present method to be brought beyond the status of proof-of-concept. Nevertheless they try to identify some steps to overcome to present limitations.

In spite of the many positive aspects outlined above, the manuscript need some improvements, listed below.

1. The Introduction is written is a rather popularized style. I would have expected a brief review of the existing literature on this topic. I understand that this is quite a new case-study of volatolomics but there is no presentation of recent papers such as ACS Nano 2020, 14, 9, 12125–12132, which might be an interesting paper to be contrasted and compared with the present one. At least, a brief account of recent papers dealing with the application of e-noses to lung diseases should be provided.

2. It is claimed that virus-infected cells produce VOCs that can be targeted for VOC-based disease detection. However, there no discussion about possible VOCs in the case of SARS CoV-2 infection. Is there any idea about the kind of biomarkes that can be related to VOCs? Could they be pneumonia biomarkers, one of the most severe consequences of SARS CoV-2 infection?

3. Though the sampling has been detailed, information about the exhaled breath is missing, such as humidity, that may affect the sensors readout. Typically, humidity in breath would require some type of filter prior to admitting the sampled breath to the sensor array; of course, one needs to make sure not to filter the analyte too. May the sampled breath of infected patients display a humidity content different from that of healthy volunteers?

4. Did the authors consider the potential effect of lung function variations on the VOCs levels in exhaled breath and, consequently, on sensor response? This is quite a general issue in e-nose applications to respiratory medicine and needs to be addressed. Data on FEV1 (forced expiratory volume in one second) and FVC (forced vital capacity) might be useful, if available, for the discrimination of volunteers.

In conclusion I believe that the paper can be published provided that the authors address the points above.

Reviewer #2: Review comments may be found on the attachment. In papers of this type, the protocols used are extremely important. If poor protocols are used, the conclusions may be deemed invalid. The reviewer believes that there is indeed an excellent case for the use of specific VOC's and their concentrations as a function of time to provide a rapid, sensitive and selective diagnosis; however, this paper fails to do so.

6. PLOS authors have the option to publish the peer review history of their article (what does this mean?). If published, this will include your full peer review and any attached files.

Reviewer #1: No

Reviewer #2: No

---

## [Author Response · Author response to Decision Letter 0]

24 Apr 2021

Dear Editor and Referees,

Thank you for the detailed response and for the reviewers’ comments. We have carefully reviewed the comments and have revised the manuscript accordingly. Our point-by-point responses are given below. 

Editor comments:

1. Please answer the concerns of reviewer #2 about the necessity to use of a sealed mask to prevent the detection of environmental contaminants, as well as the possibility that "normal" breathing patterns might not be adequate to detect endogenous VOCs.

We have answered this at two levels: First, likely due to poor wording on our part, the Referee misunderstood our aim as to conduct exhaled breath analysis, and with this in mind, had concerns regarding departure from common practice in breath sampling. However, our aim was not exhaled breath analysis, but rather smelling of the intranasal passage. We aimed for this because the nasal passage may be a primary sight of infection. Thus, we were not trying to “smell the lungs”, but rather “smell the inside of the nose”. Here, exhaled breath is in fact an unavoidable source of noise that we had to accept, but it was not our target. We have now made this very clear in the manuscript, so as to avoid misunderstandings. Second, we have added Figure 2, that clearly shows how our system is air-tight, and samples from the inside of the nose, and not environmental air. We should have likely had this figure in the original version, and hope it now clarifies things.

2. Please clarify the discrepancy between the expectation of reviewer #2 that the sensor’s conductivity should increase with the exhaled gas concentration whereas you appear to be observing the opposite.

The sensor response can be either an increase in resistance or a decrease in resistance depending on whether the gas is a reducing gas or an oxidizing gas and whether the sensor is an N type or a P type sensor (Berna, 2010): “For p type oxides, an increase in the resistance is found in the presence of reducing gases, while the resistance decreases in response to oxidizing gases; n-type oxides show opposite behavior. Examples of n type oxides are SnO2 and WO3; and a p-type oxide is CTO.” However, in the specific e-nose that we used (commercial e-nose PEN3), the conductivity usually decreases after an oxygen exchange (Baietto et al., 2010). We have now also detailed all this in the manuscript. 

3. As requested by reviewer #1, it would be important to report any information you might have about the parameters of exhaled breath (humidity) affecting the sensor's readout. 

We do not have any humidity measure. We acknowledge of course that added measures (humidity, temperature, etc) may have provided for an even better tool, and therefore throughout this manuscript we think we maintain extreme modesty regarding our claims. We never suggest that this is the best possible system for the job, we merely suggest that the eNose data alone captures meaningful variance, even in this challenging “open field” setting.

In your revised manuscript, you're encouraged (i) to add to your introduction a review of published applications of the electronic nose technology to the diagnosis of lung disease; (ii) to include a discussion of any specific biomarkers you found associated with the VOCs of SARS CoV-2.

We added to our introduction a short review of the e-nose application to lung diseases, and to COVID-19. We also commented on VOC specificity where relevant, albeit we have no data on this from our effort.

 

Reviewer 1:

Reviewer #1: The authors present a proof-of-concept of SARS CoV-2 detection with a commercially available e-nose. They present a very detailed description of in-line sampling methods at a national testing station in Tel Aviv, Israel. Many details are provided on the strategy deployed to operate in a safe environment and avoid contamination during the test of each volunteer. This is valuable contribution for future studies aimed to develop and test e-noses for the detection of such infection.

We thank the Referee for the kind words.

The data analysis was carried out correctly with machine-learning methods. The authors also provide a critical assessment of points of strength and weaknesses of both sampling method and data analysis and recognize that the main limit of their method is the level of false positives, which hinders the present method to be brought beyond the status of proof-of-concept. Nevertheless they try to identify some steps to overcome to present limitations.

We again thank the Referee for the kind words.

In spite of the many positive aspects outlined above, the manuscript need some improvements, listed below.

1. The Introduction is written is a rather popularized style. I would have expected a brief review of the existing literature on this topic. I understand that this is quite a new case-study of volatolomics but there is no presentation of recent papers such as ACS Nano 2020, 14, 9, 12125–12132, which might be an interesting paper to be contrasted and compared with the present one. At least, a brief account of recent papers dealing with the application of e-noses to lung diseases should be provided.

Consistent with this comment, we have added the following paragraph to the introduction:

Much of the eNose diagnostics effort in the literature is focused on exhaled breath analysis. Such breath sampling and analysis has standard protocols [1, 2], and has reached at achievements in cases such as identifying pneumonia [3], tuberculosis [4, 5], asthma and COPD [6], respiratory infections [7] and lung malignancies [8-10], and recently indeed for COVID-19 [11, 12]. Moreover, eNose measurements of exhaled breath may inform on non-respiratory conditions as well, such as neurodegenerative illnesses [13]. Here, however, we are not proposing exhaled breath analysis per se. Rather, we observe that the nasal passage has been implicated as a site of SARS CoV-2 infection [20-21]. Therefore, our goal is to "smell" the inner nasal passage itself. From our perspective, breath, and its associated lung-derived VOCs, are an inevitable source of noise in the nasal passage, but not our intended target. Thus, we set out to develop methods that differ from the standard breath sampling and analysis typically applied in the field.

Moreover, in further consideration of: “might be an interesting paper to be contrasted and compared with the present one”, in the discussion we now add:

We note that results to this effect have recently been obtained by others: In one study [12], the authors used a commercial metal-oxide-sensor-based eNose, albeit different from ours (Aeonose, The Aeonose Company, Zutphen, the Netherlands). They sampled oral exhaled breath from 219 participants, of which 57 were COVID-19 positive. These participants were tested at two locations, all the negative participants were tested at an outpatient clinic, and most all of the positive patients were tested at a COVID-19 nursing unit. This group obtained an ROC AUC of 0.74, compared to our ROC AUCs of 0.58 and 0.63.. In a second study [11], the authors used a proprietary nanomaterial-based hybrid sensor array eNose. This group also sampled oral exhaled breath, from 140 participants, of which 49 were COVID-19 positive, 58 were COVID-19 negative, and 33 were other lung disease participants. These participants were tested at multiple locations, with all the COVID-19 positive participants coming from the same intensive care unit at a hospital in Wuhan, China. This group obtained an ROC AUC of 0.81, again compared to our ROC AUCs of 0.58 and 0.63.. Thus, the two current published efforts that we are aware of obtained results significantly better than ours. What underlies these differences? The likely explanation is that these groups used a better device, and applied it using better sampling methods. We do not say this facetiously. That said, we would like to highlight one strength of our effort that we think is meaningful: We sampled in a naturalistic setting where, unlike in the two above examples, sources of variance where equally distributed across groups. A limitation of eNose diagnostics is that we don’t know the identity of the signal. Thus, did the above efforts detect COVID-19, or did they detect the food/laundry-detergent/any other commonality of the jointly housed COVID-19 positive participants? Here, by going directly to the very difficult end-test, namely the extremely noisy (olfaction-wise) and highly variable environment of the drive-through testing station, we provide a powerful inherently double-blind proof of concept for our hypothesis. Here, one SARS CoV-2 positive person may have just eaten a tuna-fish sandwich, another may have just brushed their teeth, and a third may have just doused themselves with their favorite perfume. Unlike an olfaction-wise homogenous group of patients in a hospital (who eat the same food, wear commonly laundered clothing, etc.), the only thing these people have in common is a later-obtained SARS CoV-2 positive test. With this noise in mind, we nevertheless obtain statistically significant classification.

2. It is claimed that virus-infected cells produce VOCs that can be targeted for VOC-based disease detection. However, there no discussion about possible VOCs in the case of SARS CoV-2 infection. Is there any idea about the kind of biomarkes that can be related to VOCs? Could they be pneumonia biomarkers, one of the most severe consequences of SARS CoV-2 infection?

This is indeed an important point. As we clearly acknowledge throughout the manuscript, we have no VOC-specific data. The only thing we can point to in this respect is the lack of contribution from sensors #1, #3, and #5, thus giving some indication on the classes of molecules not involved in this classification. That said, we do not think pneumonia was playing a role here, as these participants were mostly non-symptomatic, and indeed, our device worked better in the non-symptomatic sub-cohort.

3. Though the sampling has been detailed, information about the exhaled breath is missing, such as humidity, that may affect the sensors readout. Typically, humidity in breath would require some type of filter prior to admitting the sampled breath to the sensor array; of course, one needs to make sure not to filter the analyte too. May the sampled breath of infected patients display a humidity content different from that of healthy volunteers?

We have no data on humidity from a dedicated humidity sensor. That COVID-19 positive participants may have different humidity than COVID-19 negative participants is a keen thought, yet we think this is not the case, as humidity is primarily evident in Sensor #4, yet this sensor was not the major player in the response. 

4. Did the authors consider the potential effect of lung function variations on the VOCs levels in exhaled breath and, consequently, on sensor response? This is quite a general issue in e-nose applications to respiratory medicine and needs to be addressed. Data on FEV1 (forced expiratory volume in one second) and FVC (forced vital capacity) might be useful, if available, for the discrimination of volunteers.

We agree with the referee that lung function might be a useful predictor of COVID-19, and may affect the volume of exhaled air. We did not, and could not obtain these measures in this field-study. That said, we reiterate that here we were not measuring exhaled air in the typical sense. Participants did not exhale into our device, and our measure was not governed by the respiratory pattern. We had a constant vacuum from the nose, while participants were breathing through their mouth. We have now better clarified this in the manuscript.

In conclusion I believe that the paper can be published provided that the authors address the points above.

Than you.

 

Reviewer 2

1. Page 4 lines 8-9 state, ”When a compound interacts with the sensor, this results in an oxygen exchange that leads to decreased electrical conductivity.” This would appear to be inaccurate, as the sensor’s conductivity increases as the target gas concentration increases which may be confirmed by simply referring to the sensor manufacturer’s technical data sheet. Also, the usefulness of this paper would be greatly increased by specifically stating the VOCs in question and providing some technical data sheet information on the sensors as supplemental material.

We thank the referee for raising this point. Indeed, the sensor response can be either an increase in resistance or a decrease in resistance depending on whether the gas is a reducing gas or an oxidizing gas and whether the sensor is an N type or a P type sensor (Berna, 2010): “For p type oxides, an increase in the resistance is found in the presence of reducing gases, while the resistance decreases in response to oxidizing gases; n-type oxides show opposite behavior. Examples of n type oxides are SnO2 and WO3; and a p-type oxide is CTO.” However, in the specific e-nose that we used (commercial e-nose PEN3), the conductivity usually decreases after an oxygen exchange (Baietto et al., 2010). The following chart, supplied by the manufacturer provides the most detailed available specification of each sensor’s response:

Sensor number Sensor name Object substances for sensing Limit of detection

Sensor 1 W1C Aromatics 5 ppm

Sensor 2 W5S Ammonia and aromatic molecules 1 ppm

Sensor 3 W3C Broad-nitrogen oxide 5 ppm

Sensor 4 W6S Hydrogen 5 ppm

Sensor 5 W5C Methane, propane, and aliphatics 1 ppm

Sensor 6 W1S Broad-methane 5 ppm

Sensor 7 W1W Sulfur-containing organics 0.1 ppm

Sensor 8 W2S Broad-alcohols, broad-carbon chains 5 ppm

Sensor 9 W2W Aromatics, sulfur- and chlorine-containing organics 1 ppm

Sensor 10 W3S Methane and aliphatics 5 ppm

Per the Referee suggestion, we have now added this table to the manuscript. Beyond this very general classification, we have no detailed information on VOCs measured, as we did not use any additional analytical method (e.g., GCMS).

2. It is currently well known in the art of breath analysis detection that a unique VOC pattern is emitted from healthy individuals as well as individuals having a disease. It is also well known in breath collection protocols that environmental contamination of breath samples is a key problem that is required to be addressed in breath sample collection for maintaining the integrity of the collected sample and for facilitating accuracy of the study results.The authors teach participant’s breath samples were taken in the participant’s vehicle. Page 6, Lines 19-21 teach, “The participant was then handed the sampling valve, and instructed to hold it snugly against a nostril opening for 80 seconds. The participants were told to breathe normally through their open mouth during these 80 seconds”. The problem with this procedure set forth by the authors is that a normal breathing pattern, in my opinion, is not conducive to accurately detect endogenous VOCs, whereby, a deep breath with a long exhalation will ensure that critical VOCs retained deep within the lungs are expelled during the collection.

We completely agree with this comment, and its statement suggests that we had failed to convey a primary aspect of our effort. We were not aiming to measure exhaled breath, but rather sample the intranasal space. This is because the nasal passage is itself a primary site of infection (Brann et al., 2020; Meinhardt et al., 2021). Thus, our device was not dependent in any way on exhalation, but rather pulled a constant vacuum from the nasal passage. The connection to the nose was air-tight, such that the vacuum pulled from the nose, not from the car interior, etc. In the revised manuscript we have better clarified all this, and have added new Figure 2 that shows the air-tight sampling unit. 

4. Further, a breath sample from a single nostril presents opportunity of contaminants from entering the breath collection vessel from environmental contaminants being inhaled through the second nostril and/or the mouth, which was taught by authors to have been exposed to environmental air which inherently contain contaminants such as including, but not limited to, car exhaust, air fresheners, and/or breath from a second individual sitting in the same vehicle. Both the second nostril and the mouth should have been retained in an enclosed device such as a sealed mask to be completely sealed off from the aforesaid environmental contaminants. This would, in my opinion, been a much better protocol and would not call the results into question.

We do not dispute that we could have conducted an even better effort. The shortcoming described here stands, and there are likely more. Moreover, we now explicitly reiterate this specific shortcoming in the manuscript. In the discussion:

“our study has several weaknesses. First, our sampling strategies clearly introduce noise. For example, air from the car interior can enter through the unsampled nostril and mouth, and make its way to the sampling apparatus.”

However, we humbly submit that the possibility that we could have built a better system, does not preclude publication of our results within the very limited claims we make. The fact is that we have statistically significant classification, and we make all our raw data publicly available. We submit that although not perfect, this effort and data are valuable for the community. 

The reviewer has determined the Authors have completely disregarded standard breath collection protocols. Thus, it is my opinion that the integrity of this study has been compromised, rendering the data presented to be unsatisfactory for its intended purpose.

As noted, we acknowledge that our goal was not “breath collection” as previously practiced. We present an alternative approach, we post all the raw data, and reach at a statistically significant result. We do not present it as a medical solution, but rather merely as proof of concept for a body odor associated with COVID-19. We humbly submit that we meet this modest standard.

1. Van der Schee M, Fens N, Brinkman P, Bos L, Angelo M, Nijsen T, et al. Effect of transportation and storage using sorbent tubes of exhaled breath samples on diagnostic accuracy of electronic nose analysis. Journal of breath research. 2012;7(1):016002.

2. Scarlata S, Pennazza G, Santonico M, Pedone C, Antonelli Incalzi R. Exhaled breath analysis by electronic nose in respiratory diseases. Expert review of molecular diagnostics. 2015;15(7):933-56.

3. Schnabel R, Boumans M, Smolinska A, Stobberingh E, Kaufmann R, Roekaerts P, et al. Electronic nose analysis of exhaled breath to diagnose ventilator-associated pneumonia. Respiratory medicine. 2015;109(11):1454-9.

4. Bruins M, Rahim Z, Bos A, van de Sande WW, Endtz HP, van Belkum A. Diagnosis of active tuberculosis by e-nose analysis of exhaled air. Tuberculosis. 2013;93(2):232-8.

5. Saktiawati AM, Stienstra Y, Subronto YW, Rintiswati N, Gerritsen J-W, Oord H, et al. Sensitivity and specificity of an electronic nose in diagnosing pulmonary tuberculosis among patients with suspected tuberculosis. PLoS One. 2019;14(6):e0217963.

6. Fens N, Van der Schee M, Brinkman P, Sterk P. Exhaled breath analysis by electronic nose in airways disease. Established issues and key questions. Clinical & Experimental Allergy. 2013;43(7):705-15.

7. Joensen O, Paff T, Haarman EG, Skovgaard IM, Jensen PØ, Bjarnsholt T, et al. Exhaled breath analysis using electronic nose in cystic fibrosis and primary ciliary dyskinesia patients with chronic pulmonary infections. PLoS One. 2014;9(12):e115584.

8. Dragonieri S, Van Der Schee MP, Massaro T, Schiavulli N, Brinkman P, Pinca A, et al. An electronic nose distinguishes exhaled breath of patients with Malignant Pleural Mesothelioma from controls. Lung Cancer. 2012;75(3):326-31.

9. Machado RF, Laskowski D, Deffenderfer O, Burch T, Zheng S, Mazzone PJ, et al. Detection of lung cancer by sensor array analyses of exhaled breath. American journal of respiratory and critical care medicine. 2005;171(11):1286-91.

10. Swanson B, Fogg L, Julion W, Arrieta MT. Electronic Nose Analysis of Exhaled Breath Volatiles to Identify Lung Cancer Cases: A Systematic Review. Journal of the Association of Nurses in AIDS Care. 2020;31(1):71-9.

11. Shan B, Broza YY, Li W, Wang Y, Wu S, Liu Z, et al. Multiplexed nanomaterial-based sensor array for detection of COVID-19 in exhaled breath. ACS nano. 2020;14(9):12125-32.

12. Wintjens AG, Hintzen KF, Engelen SM, Lubbers T, Savelkoul PH, Wesseling G, et al. Applying the electronic nose for pre-operative SARS-CoV-2 screening. Surgical endoscopy. 2020:1-8.

13. Bach J-P, Gold M, Mengel D, Hattesohl A, Lubbe D, Schmid S, et al. Measuring compounds in exhaled air to detect Alzheimer's disease and Parkinson’s disease. PloS one. 2015;10(7):e0132227.

---

## [Decision Letter · Decision Letter 1]

11 May 2021

Proof of Concept for Real-Time Detection of SARS CoV-2 Infection with an Electronic Nose

PONE-D-20-27895R1

Dear Dr. Sobel,

We’re pleased to inform you that your manuscript has been judged scientifically suitable for publication and will be formally accepted for publication once it meets all outstanding technical requirements.

Kind regards,

Matthieu Louis

Academic Editor

PLOS ONE

Additional Editor Comments (optional):

Reviewers' comments:

Reviewer's Responses to Questions

**Comments to the Author**

1. If the authors have adequately addressed your comments raised in a previous round of review and you feel that this manuscript is now acceptable for publication, you may indicate that here to bypass the “Comments to the Author” section, enter your conflict of interest statement in the “Confidential to Editor” section, and submit your "Accept" recommendation.

Reviewer #1: All comments have been addressed

2. Is the manuscript technically sound, and do the data support the conclusions?

Reviewer #1: Yes

3. Has the statistical analysis been performed appropriately and rigorously? 

Reviewer #1: Yes

4. Have the authors made all data underlying the findings in their manuscript fully available?

Reviewer #1: Yes

5. Is the manuscript presented in an intelligible fashion and written in standard English?

Reviewer #1: Yes

6. Review Comments to the Author

Reviewer #1: (No Response)

7. PLOS authors have the option to publish the peer review history of their article (what does this mean?). If published, this will include your full peer review and any attached files.

Reviewer #1: No

---

## [Editor Report · Acceptance letter]

24 May 2021

PONE-D-20-27895R1 

Proof of Concept for Real-Time Detection of SARS CoV-2 Infection with an Electronic Nose 

Dear Dr. Sobel:

I'm pleased to inform you that your manuscript has been deemed suitable for publication in PLOS ONE. Congratulations! Your manuscript is now with our production department. 

Kind regards, 

on behalf of

Dr Matthieu Louis 

Academic Editor

PLOS ONE